# Bile-Based Cell-Free DNA Analysis Is a Reliable Diagnostic Tool in Pancreatobiliary Cancer

**DOI:** 10.3390/cancers13010039

**Published:** 2020-12-25

**Authors:** Caroline Driescher, Katharina Fuchs, Lena Haeberle, Wolfgang Goering, Lisa Frohn, Friederike V. Opitz, Dieter Haeussinger, Wolfram Trudo Knoefel, Verena Keitel, Irene Esposito

**Affiliations:** 1Institute of Pathology, Heinrich-Heine-University and University Hospital of Duesseldorf, 40225 Duesseldorf, Germany; silkecaroline.driescher@med.uni-duesseldorf.de (C.D.); lenajulia.haeberle@med.uni-duesseldorf.de (L.H.); wolfgang.goering@med.uni-duesseldorf.de (W.G.); lisa.frohn@med.uni-duesseldorf.de (L.F.); friederike.opitz@med.uni-duesseldorf.de (F.V.O.); 2Department of Gastroenterology, Hepatology and Infectious Diseases, Heinrich-Heine-University and University Hospital of Duesseldorf, 40225 Duesseldorf, Germany; katharina.fuchs@med.uni-duesseldorf.de (K.F.); haeussinger@med.uni-duesseldorf.de (D.H.); verena.keitel@med.uni-duesseldorf.de (V.K.); 3Department of General, Thoracic and Pediatric Surgery, Heinrich-Heine-University and University Hospital of Duesseldorf, 40225 Duesseldorf, Germany; wolframtrudo.knoefel@med.uni-duesseldorf.de

**Keywords:** pancreatic cancer, cholangiocarcinoma, cell-free DNA, liquid biopsy, ERCP, next generation sequencing

## Abstract

**Simple Summary:**

To elucidate and compare the value of plasma and bile as liquid biopsy source, cfDNA from 80 patients with pancreatobiliary cancers or non-malignant biliary obstructions was subjected to panel-based next generation sequencing (NGS). Results showed high correspondence in mutational profiles of bile-derived cfDNA and matched tissue samples, and the method proved superior to traditional plasma-based liquid biopsy techniques and with higher sensitivity than routine biomarkers such as CA19-9.

**Abstract:**

Currently available serum biomarkers for pancreatobiliary cancers lack sensitivity and specificity and ultimate diagnosis still requires invasive procedures for histological confirmation. The detection of tumor-specific genetic aberrations with utilization of cell free DNA (cfDNA) is a less invasive approach than traditional tissue biopsies; however, it has not been implemented into clinical routine. In this study, we investigated bile as a liquid biopsy source in pancreatobiliary cancers and compared its potential as cell-free DNA source to plasma. Blood (n = 37) and bile (n = 21) samples were collected from patients affected by pancreatic ductal adenocarcinoma (PDAC) and extrahepatic cholangiocarcinoma (CCA) or with non-malignant biliary obstructions (blood n = 16; bile n = 21). Panel-based next generation sequencing (NGS) and digital droplet PCR (ddPCR) were applied for tumor mutation profiling. NGS results from matched tumor tissues (n = 29) served as comparison. Sequencing of cfDNA from bile resulted in detection of 96.2% of the pathogenic tumor mutations found in matched tissue samples. On the other hand, only 31.6% of pathogenic tumor mutations found in tissue could be detected in plasma. In a direct comparison, only half of the mutations detected in bile cfDNA were concordantly detected in plasma from the same patients. Panel NGS and ddPCR displayed comparable sensitivity. In conclusion, bile is a suitable source of cfDNA for the diagnosis of pancreatobiliary cancer and performs more reliably than plasma. Although primary diagnosis still requires histologic confirmation, bile-derived cfDNA could offer an alternative if tissue sampling is not feasible and might allow less invasive disease monitoring.

## 1. Introduction

The outcome of patients with pancreatic and biliary cancers continues to be poor despite the strong efforts to improve diagnostic tools and treatment options. The poor prognosis of both malignancies relates to an intrinsic biological aggressiveness, as well as to late clinical symptoms and a lack of reliable strategies for early diagnosis [1,2]. To date, surgical resection is the only potentially curative treatment option, but only a fraction of patients are eligible for resection at the time of diagnosis, and local or distant recurrence rates continue to be high [3,4].

Suspected pancreatic ductal adenocarcinoma (PDAC) or cholangiocarcinoma (CCA) are initially evaluated by several imaging modalities with varying sensitivity [5,6]. Additionally, serum markers can be useful during the diagnostic process as well as for monitoring the treatment response, with serum carbohydrate antigen 19-9 (CA19-9) being the most extensively validated biomarker in both entities. However, CA19-9 only provides suboptimal sensitivity and specificity, especially in asymptomatic patients [7,8]. Furthermore, it is not cancer-specific and is elevated in many other hepatobiliary and gastrointestinal tumors as well as in benign biliary obstructions [9,10,11].

Histological confirmation is required for ultimate diagnosis in non-resectable patients prior to administration of a palliative chemotherapy regimen. The different modalities of invasive tissue sampling can be technically challenging and associated with severe risks, including tumor seeding along the biopsy tract, hemorrhage, organ perforation or inflammation [8,12,13]. Moreover, because of the dense desmoplastic reaction especially in PDAC, a large part of the tumor mass consists of stromal cells, giving rise to false negative results and eventually necessitating re-biopsies, which further increase the risk of complications and may delay the therapy substantially [12].

The detection of tumor-derived cell-free DNA (cfDNA) in body fluids such as plasma, urine, and saliva, represents a promising liquid biopsy approach, bypassing some of the drawbacks of classic tissue biopsies [14]. First discovered in 1948, circulating cfDNA has been shown to be elevated during different pathological processes [15]. Accordingly, the concentration of cfDNA has been shown to be higher in the blood of cancer patients compared to healthy individuals [16]. The tumor-cell derived fraction of cfDNA, so-called ctDNA (circulating tumor DNA) can be determined by the detection of tumor-specific genetic aberration, serving as a liquid-based strategy of tumor diagnosis [17]. In addition to its minimal invasiveness, this liquid-biopsy strategy offers the advantage of reflecting the heterogeneous genomic landscape more precisely than a single tissue biopsy and repeated application of the method can reveal emerging changes in the mutational profiles of tumor masses over time, which may allow for a more individualized therapeutic approach [18,19].

While blood-derived cfDNA has already been investigated as a source for detection of tumor specific alterations in PDAC and CCA in various settings, we aimed at using bile as a liquid biopsy approach for pancreatobiliary malignancies [20,21,22,23]. Biliary obstruction is often observed in patients suffering from PDAC and extrahepatic CCA and frequently requires endoscopic retrograde cholangiopancreatography (ERCP) to re-establish biliary drainage. During this procedure, bile can be collected easily without imposing any additional risks for the patient, thus representing a promising diagnostic tool in pancreatobiliary cancers.

## 2. Results

### 2.1. Detection of Pathogenic Tumor Mutations in Bile Samples

Forty-two pathogenic somatic mutations in six genes were detected in cfDNA extracted from bile from 21 tumor patients (Figure 1a and Appendix A). No mutations were found in any of the control bile samples. Across all samples from tumor patients, *KRAS* was the most frequently mutated gene, being mutated in 16 out of 17 PDAC patients and in both metastatic CCA patients. Only two patients diagnosed with localized CCA did not show any *KRAS* mutation. The mutated *KRAS* allele frequency did not show any difference between localized PDAC, CCA and metastatic PDAC (Figure 1b). The second most frequently mutated gene was *TP53* (13/21 tumor patients; 61.9%), followed by *CDKN2A* (5/21; 23.8%)*, SMAD4* (2/21; 9.5%), and *GNAS* and *BRAF* (both 1/21; 4.8%, Appendix A).

Fourteen tumor tissue samples from 21 tumor patients were available for analysis (66.7%) (Appendix A), including seven resection specimens and seven biopsies. Twenty-four of 26 (92.3%) pathogenic mutations observed in tumor tissue were detected concordantly in cfDNA from bile by software-based variant calling (Figure 1a). There were two mutations in tumor tissue that could not be detected in bile cfDNA by standard parameters. However, the *TP53* (c.994-1G>A) splice site mutation in patient 4 could be detected by manual inspection of the sequencing reads using the integrative genomics viewer (IGV) at very low allele frequency (0.4%) in the bile sample. The *GNAS* R201H mutation in the tumor tissue of patient 23 could not be found in bile even after manual inspection of sequencing reads. Therefore, manual investigation increased the overall concordance between mutations detected in bile and tumor to 96.2%. Calculation of sensitivity and specificity values of bile cfDNA sequencing in relation to tumor tissue sequencing resulted in values of 100%, respectively. In patient 23, an additional *KRAS* Q61H mutation was found in cfDNA from bile but not in FFPE-tumor DNA. Bile sampling and tumor biopsy were performed within a few days apart from each other, thereby excluding somatic evolution.

### 2.2. Detection of Pathogenic Tumor Mutations in the Plasma Samples

In total, 50 pathogenic somatic mutations in nine genes were detected in plasma cfDNA from tumor patients (Figure 2a and Appendix A). *KRAS* was also the most frequently mutated gene in these patients. Other mutations were detected in *TP53*, *SMAD4*, *CDKN2A*, *GNAS*, *BRAF*, *PIK3CA*, *CHD1,* and *IDH1* (Figure 2b). No mutations were detected in any of the control plasma samples. For 25 tumor patients, matched tumor tissue was available. Only 12/38 pathogenic tumor mutations were detected in plasma cfDNA, which resulted in a concordance rate of 31.6%. If considered separately, sequencing of plasma cfDNA from patients with localized tumors resulted in 5.9% and sequencing of plasma cfDNA from patients with metastatic tumors in 52.4% concordance rate. In 7/12 cases where no tumor tissue was available, plasma cfDNA sequencing allowed to gain information about the mutational status of the tumors (Figure 2a). The allele frequency of mutated *KRAS* in plasma cfDNA was significantly higher in patients with metastatic PDAC compared to patients with localized PDAC (*p* = 0.0166; Figure 2c).

Sequencing of plasma cfDNA in relation to tumor tissue sequencing resulted in sensitivity and specificity values of 52% and 100%, respectively, in our cohort.

### 2.3. Bile cfDNA Sequencing Performs Better than Plasma cfDNA Sequencing

From 13 patients both plasma and bile were available (Table 1), 12/25 mutations (48%) detected in bile cfDNA were concordantly detected in plasma cfDNA (Figure 3a). The concordance in patients with localized tumors was lower than that in samples obtained from patients with metastatic disease (25% vs. 75%). Comparison of *KRAS* allele frequency between bile and plasma showed significantly higher allele frequencies in bile than in plasma in patients with localized disease (*p* = 0.0159; Figure 3b). In patients with metastatic disease, the allele frequency was in most cases higher in bile samples but without significant difference. In two cases, the allele frequency was higher in plasma than in bile (Figure 3c).

Interestingly, in case 44, two pathogenic *KRAS* and two pathogenic *TP53* mutations were detected upon cfDNA sequencing from bile. The *KRAS* G12D and the *TP53* R273H mutations were present at a similar allele frequency of around 2%. *KRAS* Q61H and *TP53* R280I mutations were detected with an allele frequency of around 14%. Sequencing of plasma cfDNA from the same patient showed only the pathogenic *KRAS* and *TP53* mutations that were present at a lower allele frequency in the bile (Figure 3d).

### 2.4. CA19-9 Displays Limitations as Serum Biomarker in Our Collective

CA19-9 levels were available in 53 patients (66.3%), 12 of 35 controls (34.3%) and 41 of 45 (91.1%) tumor patients. Using the standard cut-off at our institution of 37 U/mL, 6 control patients (50%) showed elevated levels of CA19-9 (Figure 3e). On the other hand, nine tumor patients (21.4%) showed no CA19-9 elevation (Figure 3e). Only patients with metastatic PDAC showed significantly elevated levels of CA19-9 compared to the control group (*p* = 0.0082; Figure 3e). Sensitivity and specificity of CA19-9 in our patient collective was 78.6% and 50%, respectively.

### 2.5. Comparison of NGS and ddPCR

As allelic frequency of pathogenic variants is often low in liquid biopsy samples, the limit of detection of NGS-based analysis could lead to false negative results. Therefore, we performed a digital droplet PCR (ddPCR) approach for *KRAS* mutations. For comparison of both methods, the same amount of cfDNA used for NGS library preparation was used for ddPCR. cfDNA from 33/37 plasma samples and 15/21 bile samples from tumor patients were tested with ddPCR. The other samples had to be excluded from ddPCR analysis due to insufficient quantity of cfDNA or due to *KRAS* mutations in codon 61. *KRAS* allele frequency results from NGS and ddPCR were plotted together for comparison. *KRAS* allele frequency results were comparable between both methods. In all cases except one, no *KRAS* mutation was detected by ddPCR according to NGS analysis results in plasma samples (Figure 4a). In cases where *KRAS* mutations were detected, the allele frequency did not vary significantly between both methods. The same observations were made in bile samples (Figure 4b).

## 3. Discussion

Liquid biopsy is an emerging tool for the diagnosis and monitoring of neoplastic diseases. Tumor tissue remains a pre-requisite for initial diagnosis of PDAC or CCA and, to date, its availability is also essential for genomic profiling for the purpose of targeted therapy [24,25,26]. On the other hand, tissue biopsies only provide spatially and temporally limited information, which might lead to an underestimation of tumor heterogeneity [27]. Liquid biopsy methods provide advantages over tissue biopsies, such as less invasiveness, complications, and spatial limitations [14]. Plasma-based liquid biopsies have been studied extensively lately; however, apart from few exceptions, they have not been implemented into the clinical routine because of several limitations [20,21,22].

In this study, we investigated bile as a source of cfDNA and elucidated its potential role as a novel liquid biopsy source in pancreatobiliary cancers. Furthermore, we compared bile and plasma cfDNA panel-based NGS and investigated its use for the clinical routine. We found that 96% of pathogenic mutations identified by tissue sequencing could be detected in cfDNA from corresponding bile samples, yielding a sensitivity and specificity of 96.2% and 100%, respectively. In a recent prospective randomized trial, liquid-based or smear cytology of EUS-FNA obtained from solid pancreatic masses reached a sensitivity of 88% and 83.8%, respectively, for the diagnosis of pancreatic cancer [28]. Brush cytology and forceps biopsies displayed an overall sensitivity of 77.1% for the diagnosis of suspected malignant biliary strictures when obtained by ERC(P) [29]. Our results are in line with the previous studies, which reported high sensitivity and specificity of bile cfDNA analysis in gallbladder cancer and cholangiocarcinoma [30,31] and showed its value as ancillary method in the diagnosis of malignant biliary strictures [32]. This is the first study reporting on the usefulness of bile as a source of liquid biopsy in PDAC.

Sequencing of cfDNA from plasma on the other hand resulted in lower concordance and sensitivity values than bile cfDNA sequencing. Other studies have shown higher sensitivity values for plasma cfDNA sequencing, also in pancreatobiliary cancers, albeit using sequencing methods and platforms, which are difficult to integrate into routine diagnostics because of higher costs and complex pipelines [20,22,33]. Liquid biopsy may have the advantage of better representing tumor heterogeneity, overcoming the spatial limitations of tissue biopsies [27]. In patient 23, bile was collected within a day after percutaneous fine-needle biopsy of hepatic metastases and bile sequencing detected an additional pathogenic *KRAS* mutation (Q61H) compared to tissue sequencing. Since tissue and liquid biopsy were performed almost at the same time, this is unlikely to be the result of tumor progression with emergence of additional clones.

In a direct comparison, bile-based liquid biopsy performed better than plasma-based cfDNA sequencing in this study. Less than 50% of the pathogenic mutations found in bile cfDNA were also detectable in plasma samples. On the other hand, all mutations found in plasma-derived cfDNA could be confirmed via bile-based cfDNA sequencing. Sequencing of plasma cfDNA had an especially low sensitivity in tumors without distant metastases, as a consequence of lower tumor burden [34]. In addition, mutated *KRAS* allele frequency was higher in bile than in plasma, arguing for a higher “background noise” in the latter. Another important finding of the present study is the good performance of NGS compared to ddPCR, which is usually considered a more sensitive method with a limit of detection (LOD) down to 0.1% (compared to a LOD of 1% by NGS) [35].

The main obvious limitation of our study is the small number of patients, and especially of CCA cases. In addition, we do not provide a systematic comparison between different approaches to the diagnosis of pancreatobiliary malignancies. About 70% of patients with pancreatic cancer present with inoperable malignant biliary strictures requiring biliary drainage in order to improve quality of life and to allow for palliative chemotherapy treatment [36,37]. However, not all patients with pancreatobiliary cancers require therapeutic interventions for biliary drainage. In these cases, accessing bile samples for liquid biopsy purposes via endoscopic techniques would require careful assessment of risks and benefits. This includes patients with pancreatic tumors located in the body or tail of the pancreas, who account for approximately one-third of the PDAC patients [38]. In our collective, 88.6% of patients presented with tumors located in the pancreatic head, whereas only four patients (11.4%) had tumors located in the pancreatic body/tail and one of them with a pancreatic body tumor required biliary drainage. Therefore, we cannot elucidate whether the tumor localization affects the detection of tumor mutations in bile cfDNA. This aspect should be taken into account in larger study cohorts.

A potential source of false positive results in plasma-based liquid biopsy is the common age-related phenomenon of clonal hematopoiesis, in which hematopoietic cells accumulate non-malignant mutations and cause genetically distinct subpopulations of white blood cells [39,40]. As the majority of cfDNA in plasma originates from peripheral blood cells and clonal hematopoiesis is highly prevalent in both healthy individuals and cancer patients, this potential mutational background noise should be taken into account when interpreting liquid biopsy results [41,42]. Therefore, in cases where no tumor tissue was available for comparison, false positive results of plasma cfDNA analysis cannot be completely excluded; on the other hand, no pathogenic variants were found in any of the controls. Mutations related to clonal hematopoiesis might play a minor role when interpreting results from bile-derived cfDNA because of the different composition of cells in bile and plasma. However, further investigation is needed to confirm this assumption.

Despite these limitations, we show that bile-based cfDNA analysis by NGS proves as a feasible and beneficial ancillary diagnostic tool for pancreatobiliary cancers, especially in those cases where a tissue-based diagnosis is not possible displaying higher sensitivity and specificity than plasma sequencing for the detection of tumor specific aberrations. In the future, bile-based cfDNA analysis may improve early diagnosis of pancreatic cancer, if used in a screening setup for patients at risk. Furthermore, it may enable disease- and therapy-monitoring when used as a follow-up method.

## 4. Materials and Methods

### 4.1. Patients’ Collective

Bile was collected during interventional ERCPs from patients with pancreatic ductal adenocarcinoma (PDAC) (n = 17) or cholangiocarcinoma (CCA) (n = 4). Blood samples were taken from 29 PDAC and 8 CCA patients. Matched tumor tissues (n = 29) were used as control. Control blood (n = 16) and bile samples (n = 21) were collected from non-tumor patients with an indication for an interventional ERCP (Table 1 and Appendix A). All bile and blood samples were collected between October 2018 and July 2020.

### 4.2. Blood and Bile Samples and cfDNA Isolation

Venous blood was collected in cell-Free DNA BCTs^®^ (Streck, Nebraska, US) following manufacturer’s instructions and bile was withdrawn through a catheter from the biliary tree during ERCP. The amount of bile varied between 1 mL and 5 mL and all volumes were sufficient for cfDNA isolation. Prior to cfDNA isolation, plasma was separated from 10 mL whole blood by centrifugation at 600 g for 20 min and ERCP-obtained bile was filtered through a 70-µM filter to remove larger debris. Plasma and bile samples were then centrifuged at 1600 g for 10 min at room temperature to remove any remaining cells. cfDNA was extracted using QiAamp circulating nucleic acid kit (Qiagen, Hilden, Germany) and quantified by a custom-made qPCR assay (Primer for: 5′AAACGCCAATCCTGAGTGTC-3′; Primer rev: 5′CATAGCTCCTCCGATTCCAT-3′).

### 4.3. DNA Isolation from Tumor Tissue

Tumor DNA was extracted from formalin-fixed, paraffin-embedded (FFPE) tissue samples using the GeneRead DNA FFPE kit (Qiagen, Hilden, Germany) and quantified as described above.

### 4.4. DNA Next Generation Sequencing (NGS)

cfDNA or tissue-derived DNA were amplified using Ion AmpliSeq™ Cancer Hotspot Panel v2 and processed using Ion AmpliSeq™ Library Kit 2.0 with Ion Xpress™ Barcode Adapters. Libraries were quantified using Ion Library TaqMan™ Quantitation Kit and pooled following manufacturer’s instruction. Pooled libraries were subjected to sequencing using Ion 520™ & Ion 530™ Kit-OT2 on an Ion S5™ System.

Primary data analyses were performed by S5 Torrent Server VM Variant calling and variant annotation were facilitated by the Ion Reporter Software (version 5.12). The standard parameters for variant calling analysis were set as following: minimum allele frequency 1%, minimum coverage 500, Phred-score >30. Detected variants were inspected in detail and their clinical significance was determined by using different tools and databases (e.g., Integrative Genomics Viewer (IGV), ClinVar, OncoKB, COSMIC, FATHMM, PoyPhen-2). All reagents and software were from Thermo Fisher Scientific (Darmstadt, Germany).

Mutations called from bile cfDNA sequencing were compared to mutations from matched tumor tissue sequencing and concordance was determined for all mutations across the cohort. Additionally, mutations in bile cfDNA were compared to mutations detected in plasma cfDNA and concordance was determined as mentioned above.

The allele frequency of mutations was determined per gene and the mutations were compared using an oncoprint.

### 4.5. Digital Droplet PCR (ddPCR)

Mutant *KRAS* variants in cfDNA from bile and plasma were also analyzed using a QX200 Droplet Digital PCR system. The same amount of cfDNA used for sequencing was tested with the ddPCR^TM^ KRAS Screening Multiplex Kit. The reaction mixture was prepared according to the manufacturer’s protocol and subsequently transferred to a QX200 droplet reader for fluorescence measurement. Droplets were scored positive or negative based on their fluorescence intensity. The ratio of mutated fragments (fractional abundance) was calculated by QuantaSoft (ver. 1.7) software based on the Poisson distribution. All reagents and software were from Bio-Rad (Hercules, CA, USA).

### 4.6. Sensitivity and Specificity

Sensitivity and specificity were calculated per patient by comparing the pathogenic mutations found in cfDNA to tumor DNA mutations [43]. cfDNA mutations were defined as true positive if they were concordant with those found in DNA extracted from tissue.

### 4.7. Statistical Analysis

Mann–Whitney *U* or Kruskal–Wallis test with multiple comparisons were performed to determine the significance. *p*-values < 0.05 were considered statistically significant. Correlations were calculated with Spearman’s rank correlation coefficient. All statistical analyses were calculated using GraphPad Prism (ver. 8.0.2).

## 5. Conclusions

In the present study, we show that bile-based liquid biopsy by NGS might prove beneficial as an ancillary diagnostic tool for pancreatobiliary cancers, especially in those cases where a tissue-based diagnosis is not possible. Furthermore, liquid biopsies obtained from bile may improve disease monitoring.

## Figures and Tables

**Figure 1 cancers-13-00039-f001:**
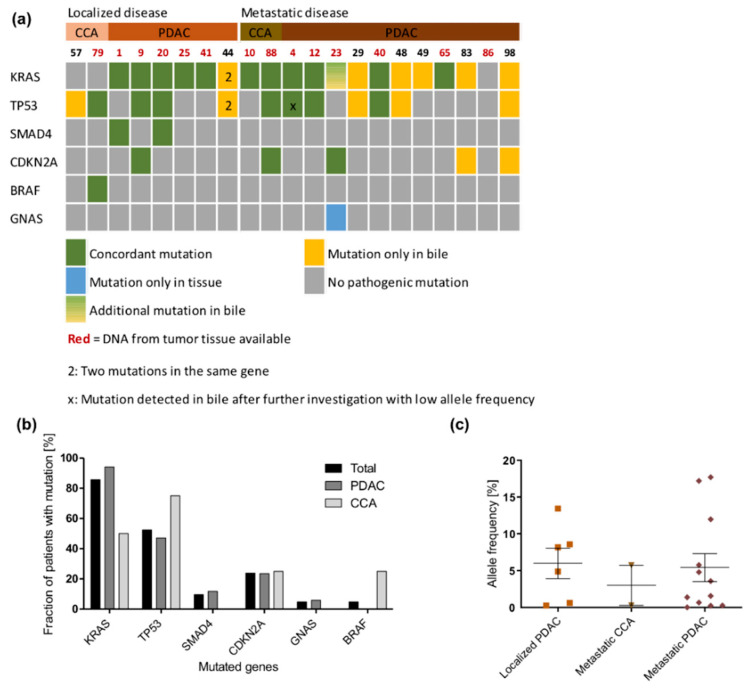
Next generation sequencing (NGS) analysis in bile samples: (**a**) Oncoprint showing mutation occurrence of six most frequently mutated genes across all patients. Patients are divided into those with localized and those with metastatic tumors. Concordant mutations are displayed in green, while mutations only detected in tissue or bile are displayed in blue or yellow, respectively. (**b**) Distribution of mutations within the bile samples. (**c**) *KRAS* allele frequency in bile samples. Patients with localized cholangiocarcinoma (CCA) were excluded, as no *KRAS* mutation was detected in the bile samples from these patients.

**Figure 2 cancers-13-00039-f002:**
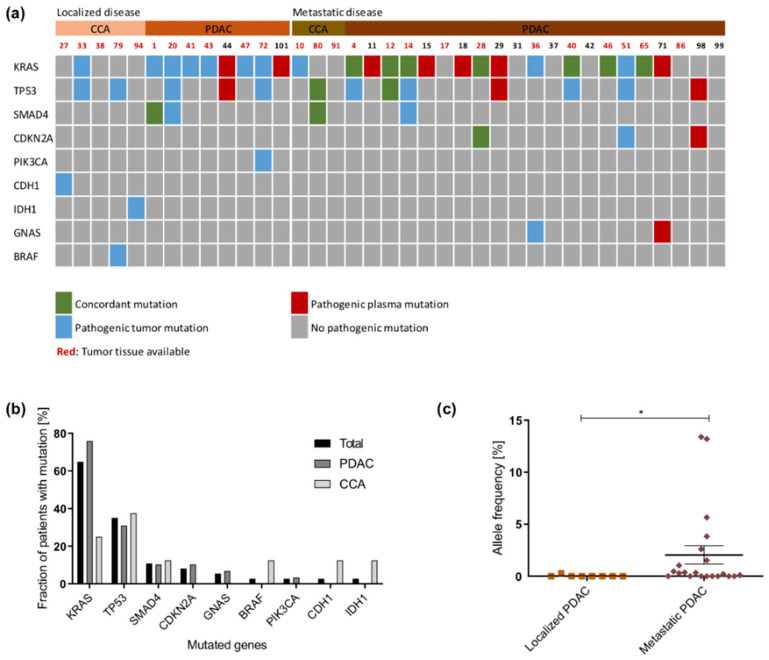
NGS analysis in plasma samples: (**a**) Oncoprint showing mutation occurrence of nine most frequently mutated genes across all patients. Patients are divided into those with localized and those with metastatic tumors. Concordant mutations are displayed in green, while mutations only detected in tissue or plasma are displayed in blue or red, respectively. (**b**) Distribution of mutations across the plasma samples. (**c**) *KRAS* allele frequency in plasma samples. Samples from CCA patients were excluded, as no *KRAS* mutation was detected in the plasma samples from these patients. *: *p* < 0.05.

**Figure 3 cancers-13-00039-f003:**
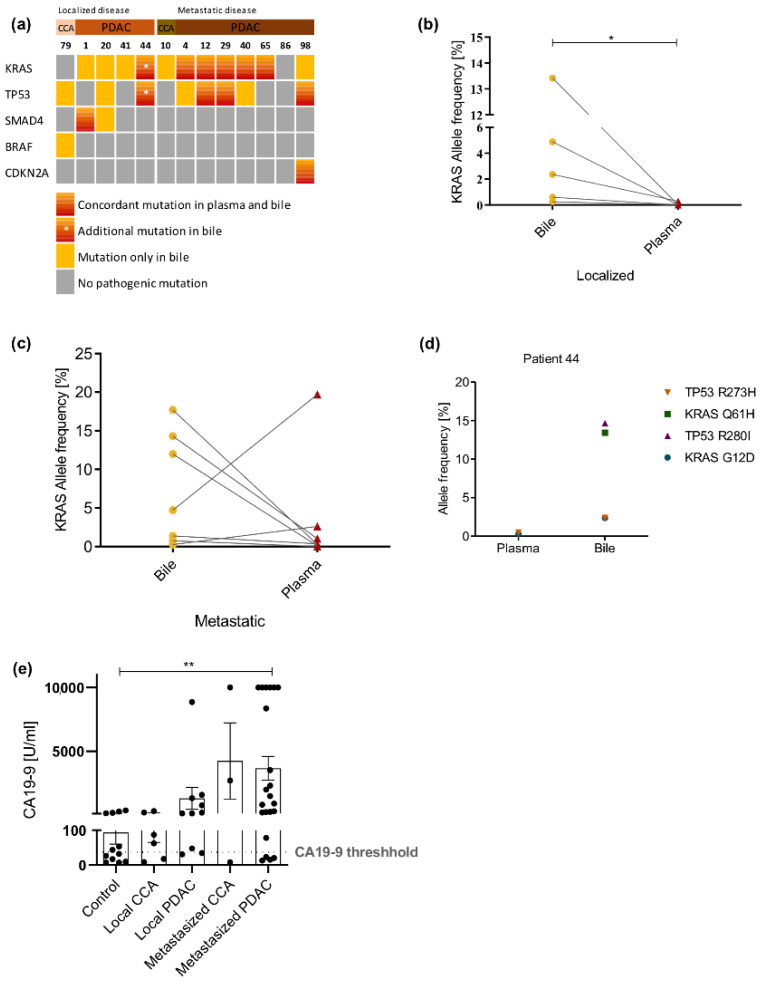
Comparison of NGS results between bile and plasma. (**a**) Oncoprint showing comparison of mutation occurrence in five top genes across all patients in blood and bile cfDNA samples. Patients are divided into controls, patients with localized and patients with metastatic tumors. *KRAS* allele frequency in bile and plasma cfDNA in patients with localized (**b**) and metastatic disease (**c**). (**d**) *KRAS* and *TP53* allele frequencies of patient 44 in plasma and bile. (**e**) CA19-9 serum levels of patients within our cohort. *: *p* < 0.05; **: *p* < 0.005.

**Figure 4 cancers-13-00039-f004:**
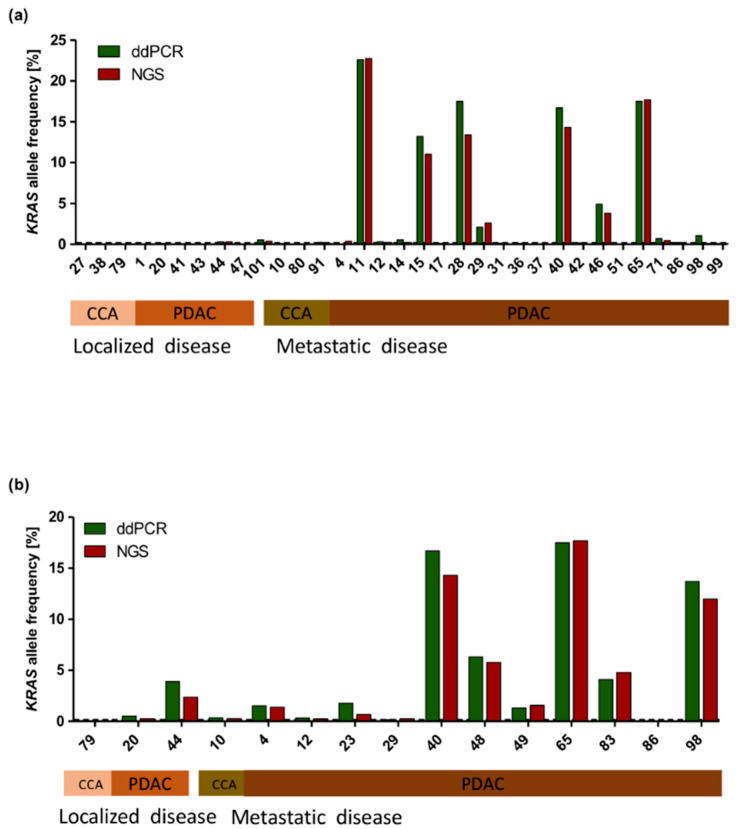
Comparison of *KRAS* allele frequency measured by NGS vs. ddPCR. *KRAS* allele frequency displayed in tumor samples measured by ddPCR (green) and NGS (red) in plasma (**a**) and bile samples (**b**). Some patients had to be excluded from analysis because of insufficient quantity of DNA or because of *KRAS* mutations in codon 61.

**Table 1 cancers-13-00039-t001:** Overview of patient collective.

Diagnosis	N	Age (Mean)	Gender ♂:♀	Bile Samples	Plasma Samples	Tissues	CA19-9 (U/mL)
**Localized disease**	**16**	**66.5**	**8:8**	**9**	**14**	**13**	**856.1**
CCA	6	62.5	4:2	2	5	5	112.0
PDAC	10	70.8	4:6	6	8	8	1302.6
**Metastatic disease**	**29**	**64.6**	**18:11**	**13**	**24**	**16**	**3725.1**
CCA	4	58.0	2:2	2	3	4	4235.5
PDAC	25	65.6	16:9	11	21	12	3655.6
**Control**	**35**	**63.7**	**20:15**	**23**	**14**	**-**	**93.5**
CBD obstruction	10	69.5	7:3	10	0	-	126.7
Choledocholithiasis	4	80.0	0:4	4	1	-	-
Chronic pancreatitis	7	50.1	5:2	3	4	-	53.1
IPMN	6	72.5	1:5	1	5	-	17.7
PSC	5	49.0	3:2	5	1	-	130.0
Pseudocyst	5	63.0	4:1	1	4	-	6.7

CCA: cholangiocarcinoma; PDAC: pancreatic ductal adenocarcinoma; CBD: common bile duct; IPMN: intraductal papillary mucinous neoplasm; PSC: primary sclerosing cholangitis. Bold: the sum of the sections.

## Data Availability

The data presented in this study are available on request from the corresponding author. The data are not publicly available due to restrictions eg privacy or ethical.

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
