# Peer review of "Bile-Based Cell-Free DNA Analysis Is a Reliable Diagnostic Tool in Pancreatobiliary Cancer"

_cancers, 2020, doi:10.3390/cancers13010039_

Round 1

Reviewer 1 Report

  1. There is a mistake in the second paragraph title:
    ".....mutations in the bile samples"
    rather than ".....mutations in the plasma samples".
  2. In figure 3d: invert bile with plasma position on the graph abscissae axis, for to be in line with the other charts.

The study is very interesting and innovative for the first time was analyzed the comparison between mutations identified in bile compared to the plasma in the same patient. It would be interesting to also study patients with pancreatitis with bile diagnosis.

Finally, it would be interesting to understand also if alterations in DNA methylation could give more significant results, but this is another paper.

Author Response

  1. There is a mistake in the second paragraph title: ".....mutations in the bile samples"
    rather than ".....mutations in the plasma samples".

The mistake in line 116 has been corrected.

  1. In figure 3d: invert bile with plasma position on the graph abscissae axis, for to be in line with the other charts.

We changed the graph titles in figure 3d.

  1. The study is very interesting and innovative for the first time was analyzed the comparison between mutations identified in bile compared to the plasma in the same patient. It would be interesting to also study patients with pancreatitis with bile diagnosis.

As shown in table 1, we have included seven patients with chronic pancreatitis in our collective and received bile from four of these patients. In none of these patients a pathogenic mutation was detected in cfDNA sequencing, indicating a high specificity of the method. These observations will be validated in the future with larger patient cohorts.

  1. Finally, it would be interesting to understand also if alterations in DNA methylation could give more significant results, but this is another paper.

This is definitely an interesting point, which we are willing to take into account in further studies.

Reviewer 2 Report

The manuscript entitled ‘Bile-based cell-free DNA analysis is a reliable diagnostic tool in pancreatobiliary cancer’ by Driescher etal shows that analysis of bile fluid cf-DNA can better predict the incidence PDAC compared to plasma cf-DNA analysis. Lack of early detection combined with the heterogeneity of the tumor niche makes PDAC one of the most difficult cancers to treat and majority of the patients succumb to the terrible disease. Thus, a minimally invasive and reliable diagnosis tool would be ideal for better prediction and treatment strategy. The authors conducted studies on matched plasma samples, tumor biopsies and bile fluid to confirm the disease. Overall, it is a sound manuscript but there are some concerns that the authors could address.

Specific comments:

  1. The authors mention the analysis of cf-DNA in both bile and plasma; it would be great if the authors can address how they ruled out the possibility of mutations originated from clonal hematopoiesis, which can occur during normal aging; a comparative study of the mutational status of the identified genes in the circulating WBCs would be ideal to rule out this possibility.
  2. The authors should include the site of the tumor (anatomical location – head, neck or tail of pancreas), if available, and mention whether this affects the detection of the mutations in bile vs. plasma, this would be helpful to rule out if the location affects the diagnosis from bile samples.

Minor comments:

Under results section, 2.1 and 2.2, the authors have provided same subtitles for two different analyses; please change the subtitle in 2.2 to ‘Detection of pathogenic tumor mutations in plasma samples’

Author Response

  1. The authors mention the analysis of cf-DNA in both bile and plasma; it would be great if the authors can address how they ruled out the possibility of mutations originated from clonal hematopoiesis, which can occur during normal aging; a comparative study of the mutational status of the identified genes in the circulating WBCs would be ideal to rule out this possibility.

This is an interesting point that we did not consider before and might be relevant especially for the interpretation of the results of plasma cfDNA analysis. We hypothesize that due to the different composition of cells in bile and plasma, clonal hematopoiesis mutations might play a minor role when interpreting results from bile-derived cfDNA. However, further investigation would be needed to confirm this assumption. We have now included a section about WBCs and clonal hematopoiesis into our discussion (lines 242-251).

  1. The authors should include the site of the tumor (anatomical location – head, neck or tail of pancreas), if available, and mention whether this affects the detection of the mutations in bile vs. plasma, this would be helpful to rule out if the location affects the diagnosis from bile samples.

We included the information about tumor localizations in supplementary table S1 and inserted a section in the discussion (lines 234-240). Due to our small collective, we cannot rule out whether the tumor location affects the diagnosis as 88.6% of our tumor patients had tumors in the pancreatic head. Overall, the majority (two-thirds) of PDAC is located in the pancreatic head, which cause more often the indication for biliary drainage. However, this will be an aspect that needs to be evaluated in a larger study cohort. 

  1. Under results section, 2.1 and 2.2, the authors have provided same subtitles for two different analyses; please change the subtitle in 2.2 to ‘Detection of pathogenic tumor mutations in plasma samples’

This mistake in line 116 has been corrected in the revised version of the manuscript.

Reviewer 3 Report

This report compared the value of plasma and bile as liquid biopsy source. And concluded bile is a suitable source of cfDNA for diagnosis of pancreatobiliary cancer and performs more reliably than plasma.

Pancreatobiliary cancers are difficult to obtain the tumor tissue because of anatomical difficulty. And plasma liquid biopsy have the problem of low sensitivity. Therefore, this report is valuable that indicated the bile as high sensitivity without tumor biopsy.
I have some minor comments.

Line 116
2.2 Detection of pathogenic tumor mutations in the bile samples
"bile samples" was erratum? (plasma is correct?)

Usually pancreatic body/tail cancer do not invade to bile duct and do not cause obstractive jaudice. Is subjects of this report include pancreatic body/tail cancers? Please clarify the anatomical site of tumors in pancreatic cancer patients. If this report did not included pancreatic body/tail cancers, it would be one of the weak point of Bile-based cell-free DNA analysis.

In method section, you should describe about the amount of bile juice needed for liquid biopsy for bile juice.

In discussion, line199-203, please clarify what kind of sensitivity did you discuss, sensitivity of pathological diagnoses, or sensitivity of genetic test (detect of KRAS mt).

Which is the exit strategy of bile-based cfDNA, diagnosis (early cancer detection) or genetic screening and monitoring for targeted therapy?

Author Response

  1. Detection of pathogenic tumor mutations in the bile samples
    "bile samples" was erratum? (plasma is correct?)

This mistake in line 116 has been corrected in the revised version of the manuscript.

  1. Usually pancreatic body/tail cancer do not invade to bile duct and do not cause obstructive jaundice. Is subjects of this report include pancreatic body/tail cancers? Please clarify the anatomical site of tumors in pancreatic cancer patients. If this report did not included pancreatic body/tail cancers, it would be one of the weak point of Bile-based cell-free DNA analysis.

We included the information about tumor localizations in supplementary table S1 and inserted a section in the discussion (lines 234-240). Due to our small collective, we cannot rule out whether the tumor location affects the diagnosis as 88.6% of our tumor patients had tumors in the pancreatic head. Overall, the majority (two-thirds) of PDAC is located in the pancreatic head, which cause more often the indication for biliary drainage. However, we agree with the reviewer that this aspect needs to be evaluated in a larger study cohort. 

  1. In method section, you should describe about the amount of bile juice needed for liquid biopsy for bile juice.

This information has now been included in the revised version of the manuscript (lines 270-280). Bile volume ranged between 1 and 5 ml and all volumes were sufficient for cfDNA isolation.

  1. In discussion, line199-203, please clarify what kind of sensitivity did you discuss, sensitivity of pathological diagnoses, or sensitivity of genetic test (detect of KRAS mt).

The meaning of the term sensitivity has now been clarified (lines 201-202) and it was referred to the diagnosis of pancreatic cancer.

  1. Which is the exit strategy of bile-based cfDNA, diagnosis (early cancer detection) or genetic screening and monitoring for targeted therapy?

In this manuscript, we have assessed the role of bile-based cfDNA analysis as ancillary diagnostic tool, as guidelines require histologic conformation for ultimate diagnosis. However, as you suggested, other applications can be envisaged, as now discussed in lines 254-256.

Round 2

Reviewer 2 Report

The authors have addressed my concerns and I have no further comments. The manuscript can be accepted for publication.